# Online Signature Verification Based on a Single Template via Elastic Curve Matching

**DOI:** 10.3390/s19224858

**Published:** 2019-11-07

**Authors:** Huacheng Hu, Jianbin Zheng, Enqi Zhan, Jing Tang

**Affiliations:** 1School of Information Engineering, Wuhan University of Technology, Wuhan 430070, China; honyeal@whut.edu.cn (H.H.); jbzheng@whut.edu.cn (J.Z.); 2Hubei Collaborative Innovation Center for High-Efficient Utilization of Solar Energy, Hubei University of Technology, Wuhan 430068, China; mimitang85119@163.com

**Keywords:** curve similarity, curve similarity model, curve similarity transformation, similarity distance, segmentation matching, evolutionary computation

## Abstract

Person verification using online handwritten signatures is one of the most widely researched behavior-biometrics. Many signature verification systems typically require five, ten, or even more signatures for an enrolled user to provide an accurate verification of the claimed identity. To mitigate this drawback, this paper proposes a new elastic curve matching using only one reference signature, which we have named the curve similarity model (CSM). In the CSM, we give a new definition of curve similarity and its calculation method. We use evolutionary computation (EC) to search for the optimal matching between two curves under different similarity transformations, so as to obtain the similarity distance between two curves. Referring to the geometric similarity property, curve similarity can realize translation, stretching and rotation transformation between curves, thus adapting to the inconsistency of signature size, position and rotation angle in signature curves. In the matching process of signature curves, we design a sectional optimal matching algorithm. On this basis, for each section, we develop a new consistent and discriminative fusion feature extraction for identifying the similarity of signature curves. The experimental results show that our system achieves the same performance with five samples assessed with multiple state-of-the-art automatic signature verifiers and multiple datasets. Furthermore, it suggests that our system, with a single reference signature, is capable of achieving a similar performance to other systems with up to five signatures trained.

## 1. Introduction

Biometric authentication has always been a field of primary concern in the security application field [1,2]. Person authentication or verification using handwritten signatures is one of the most widely researched behavior-biometrics and the most popular method for identity verification [3]. Usually, signature verification systems can be divided into two categories, namely, off-line and on-line systems, which have a significant difference. Dynamic signatures are too difficult to imitate and forge, even for skilled forgers [4] because they are unique and consistent for a given period. Compared with off-line signatures [5], online signatures are more robust and gain a higher level of security by monitoring dynamic features like time series of position trajectories, pressure, altitude, and azimuth. There is a tendency to recover online signatures from offline signature images [6].

Online signature verification can basically be viewed as a problem of similarity discrimination, whereby a decision must be made about whether a given online signature corresponds to the claimed identity or not. In a signature verification system, we compare the features of a test signature against those from a set of genuine signatures of an enrolled user, which can be called reference signatures or template signatures. By stable and discriminative feature extraction and selection, there are two approaches to identify the authenticity of a signature, which can be called the function approach and the parameter approach.

Signature verification methods based on the parameter approach include the statistical classification [7,8], neural network [9], support vector machine (SVM) [10], Bayesian decision [11] and features cluster [12], where some global or local features derived from the original signature signal, e.g., average speed, pressure, the number of strokes, etc., constitute signature feature patterns or feature vectors.

The signature verification system based on a function considers each signature signal as a function of time and verifies the signer by comparing the reference signature with the test signature directly. Usually, matching procedures or special function parameter calculations are a need between signatures, requiring more time and space. The common approaches include Dynamic Time Wrapping (DTW) [13], its improved version [14,15,16,17,18,19] and the hidden Markov model (HMM) [20,21].

Bonus template matching approaches are considered, and a longest common subsequences (LCSS) combined elastic distance metrics is also used [22]. A discrete cosine transform (DCT) [23] has been applied to 44 time signals. A multi-section vector quantization (VQ) approach [24] has been suggested where all signatures are represented by vectors of the same length. Similar methods such as the Fourier description [25], wavelet packet and discrete wavelet transform (DWT) [26] have been presented too.

A function-based system utilizes all original information about the signature, and shows better performance than parameter-based systems. Nowadays, fusion for improving verification accuracy has become a promising trend, and a combination of parametric approaches and functional approaches is often adopted in literature [16,17,18,27].

### 1.1. Related Work

Although a signature can show individual behavior features, it is more unstable and diverse than other biometric verification technologies such as fingerprint recognition, iris recognition, face recognition and so on. Due to changes in the internal and external environment, there are fluctuations in the size, location and rotation angle of signatures with the same signer at different input times. In addition, signatures will not maintain high consistency for a long time as writing habits and the external environment change. As a result, two repetitions of a signature from the same writer never have an identical appearance [28]. Each person can even have several signatures of diverse styles, and one style of signature is obviously not suitable for the verification of another style of signature. Of course, the style variability also makes signature verification better for privacy protection than fixed biometric recognition technology.

In a traditional signature verification system, a large number of samples are mandatory when building a reliable statistical classifier and many algorithms even also require skilled forgery samples [29]. Figuring out a stable signature region is also a hot topic in recent research. Similarly [19,29], extraction of a stable signature region also depends on availabiliy of a large number of training signatures. In practice, it is often impractical to obtain various signature samples from a signer, which limits the applicability of a signature verification system.

How to reduce the enrollment signature size is a crucial issue. Another problem is how to reduce the differences between different signatures, that is, the problem of signature alignment is also a key problem to be solved.

It is the most widely used and recommended method for size alignment by max-min normalization [30]. Some template matching methods, such as DTW [14], LCSS [22] and so on, also apply for alignment. Recently, alignment methods based on Gaussian mixture model (GMM) have been developed [17], but training a Gauss model requires a large number of samples.

For selecting effective reference signatures, the intra-class variation of genuine signatures can be quantified with a correlation-based criterion which detects and recovers non-linear time distortions in different specimens as described in [31].

A single reference signature system (SRSS) for training with only a single reference signature has been proposed in [28], which followed the strategy of duplicating the reference signature to enlarge the training set. In this work, the strategy consists of duplicating the given signature a number of times and training an automatic signature verifier with each of the resulting signatures and the duplication scheme is based on a sigma lognormal decomposition of the reference signature.

Nevertheless, in a real situation, it is sometimes difficult to obtain enough signatures from a signer, especially in commercial applications and forensic covers. Therefore, this paper discusses the model and method of designing an automatic signature verification system using only one real reference signature per enrolled signer. Moreover, in this study, it is vitally important to effectively align the test signature to the reference signature for verification in order to cut down the influence of fluctuations caused by variances of size, location and rotation angle, which may deteriorate the performance of verification.

The signature trajectory can be viewed as a 2D/3D curve. The similarity between two signatures can be measured by curve similarity [32]. Curve similarity is a major category of similarity measure and a large number of similarity problems can be transformed or abstract into curve similarity problems.

Measuring curve similarity is a common method for curve matching. The curves are usually assumed to be represented as polygonal chains in the plane and to be measured by distance such as DTW or Fréchet distance as in [33]. The Fréchet distance, which relies on fewer features, can be applied for signature verification as proposed in [33].

By computing cumulative distance, DTW provides normalization and alignment as a computational technique to determine the best match between two curves, which might produce different sample points. The Fréchet distance belongs to a general class of distance measures that are sometimes called “dog-man” distances and is a max measure which is outlined in terms of the maximum leash length over a parameterization. However, two classical curve similarity measures are sensitive to data anomaly points and cannot adapt to changes in the translation and scaling of the curve.

### 1.2. This Paper

Our goal of this paper is to design an automatic signature verification system for a SRSS. To this end, a new curve similarity measure model and calculation method has been established, which we call the curve similarity model (CSM). The curve similarity draw lessons from geometric similarity, and can be adapted to various transformations such as translation, scaling and rotation, and can better be adapted to the inconsistency of signatures such as signature size, position and rotation angle in the signature curve.

The procedure presented in this paper considers a rigorous and adaptive CSM to build a robust SRSS. Therefore, we completed the exploratory work reported in [32], proposed a continuous and discrete curve similarity model based on transformation, and accomplished the curve optimal matching calculation based on evolutionary computation (EC) [32]. Based on the characteristics of the SRSS, a differentiated fusion feature named local similarity score (LSC) is designed for the difference calculation between two signatures.

The paper is organized as follows: Section 2 introduces the relevant definitions of CSM. Section 3 describes the proposed curve similarity calculation method and process. The fourth section describes the optimal sectional matching of signature curves and local matching feature extraction for SRSS. The experimental results will be presented and discussed in Section 3. Conclusions are drawn in the last section.

## 2. Model and Method

### 2.1. Curve Similarity Model

#### 2.1.1. Original Definition

One curve is typically represented by a function. A definition must be provided to study the problem of the curve similarity. In geometry, there is a strict definition for shape similarity, which is an accurate similarity. In engineering applications, due to the large number of error factors, the definition of fuzzy curve similarity is adopted. Taking a 2D plane curve as an example, a kind of curve similarity is defined as follows:

**Definition** **1**([32])**.**
*Given functions f*_1_*(x) and f*_2_*(x),*
$d(f_{1},f_{2})=\int_{C_{1}}^{C_{2}}\left|f_{1}(x)-f_{2}(x)\right|dx$
*is the distance between two functions, and also known as the function similarity distance or the curve similarity distance, where [C_1_, C_2_] is the function definition domain or the definition domain*.

**Definition** **2**([32])**.**
*For a given threshold ε, if d(f_1_,f_2_) < ε, then f_1_(x) and f_2_(x) are similar, otherwise they are not.*


As mentioned above in the definitions of curve similarity distance (CSD) and curve similarity, the two functions have the same definition domains, that is to say, two curves must be aligned first, which is very limited in practical application. In most cases, the definition domains of two functions are different, and it is necessary to perform a truncation, translation, stretching, or even rotation transformation on a function to calculate the similarity distance.

As shown in Figure 1, given a curve *L*, the curves *L*_1_ to *L*_3_ are new curves obtained after applying different transformations such as translation or stretching. If the above calculation method is adopted, the distances between *L*_1_ to *L*_3_ and *L* are different. Similar problems are available in the calculation of DTW and Fréchet distances of the curves.

From the perspective of transformation, the above several curves are similar, and the distance is 0. Therefore, when to measure the similarity between curves, the transformation of the curve should be taken into consideration. Firstly, a curve similarity distance definition based on curve transformation may be suggested.

#### 2.1.2. Improvement Definition

**Definition** **3.**
*Given functions f*
_1_
*(x) defined on [R*
_1_
*, R*
_2_
*] and f*
_2_
*(x) defined on [C*
_1_
*, C*
_2_
*], and make transform*
$T\to {f^{\prime }}_{1}(x)=k\cdot f_{1}((x-b)/a)-h$
*, then the minimum distance of all the matching distances between f’_1_(x) and f_2_(x) with different transformation T is as follows:*
(1)$$\begin{array}{ll} dis(a,b,k,h) & =\int_{aR_{1}+b}^{aR_{2}+b}\left|T(f_{1}(x))-f_{2}(x)\right|dx\\ & =\int_{aR_{1}+b}^{aR_{2}+b}\left|{f^{\prime }}_{1}(x)-f_{2}(x)\right|dx\\ & =\int_{aR_{1}+b}^{aR_{2}+b}\left|k\cdot f_{1}((x-b)/a)-h-f_{2}(x)\right|dx\\ & =\int_{R_{1}}^{R_{2}}a\cdot \left|k\cdot f_{1}(t)-h-f_{2}(a\cdot t+b)\right|dt\begin{array}{cc} & t=(x-b)/a \end{array}\\ & =\int_{R_{1}}^{R_{2}}a\cdot \left|k\cdot f_{1}(x)-h-f_{2}(a\cdot x+b)\right|dx \end{array}$$
(2)$$d(f_{1},f_{2})=d({f^{\prime }}_{1},f_{2})=\min\left\{dis(a,b,k,h)\right\}$$
*which is called the curve similarity distance of f_2_(x) to f_1_(x) under the transformation T, where f_1_(x) is called the reference function or the reference curve, f_2_(x) is called the comparison function or the comparison curve, f’_1_(x) is called the transform function or the transform curve, T is called a function similarity transformation or a curve similarity transformation (CST), and dis(a,b,k,h) is called the distance of the two curves under the curve similarity transformation T.*


Obviously, the distance between the two curves is different under different similarity transformations *T*, and the curve similarity distance (CSD) is the distance after the optimal matching of the reference curve for the comparison curve. After the curve similarity transformation, the curve similarity distance is denoted by min{*dis(a,b,k,h)*}.

Once the optimal matching of the curves is obtained, the corresponding curve similarity distance can be obtained, as shown in Figure 2.

From Figure 2, *a*, *k* are translational transformations in the horizontal and vertical directions, respectively, and *b*, *h* are scaling transformations in the horizontal and vertical directions, respectively.

Therefore, the similarity of curves based on curve similarity transformation is defined as follows:

**Definition** **4.**
*Given a reference curve f_1_(x) and a comparison curve f_2_(x), for a given threshold ε, if d(f_1,_f_2_) ≤ ε, then the curve f_2_(x) is called similar to curve f_1_(x), and vice versa.*


#### 2.1.3. Discrete Definition

In engineering applications, the expression of the curve is therefore difficult to obtain, and can only be represented by an implicit function. By sampling, a continuous curve can be represented by a set of discrete ordered points. In this way, it is a common problem to measure the similarity of the two curves, which is to determine the similarity between two discrete ordered sets. As a result, a discrete curve similarity definition is needed. Similarly, the definition of discrete curve similarity transformation is given by reference to planar image transformation.

**Definition** **5.***Given one discrete curve F_A_ = {(x_1_, y_1_), (x_2_, y_2_)…(x_m_, y_m_)}, and*(3)$$T=\left[\begin{array}{ccc} a & 0 & 0\\ 0 & k & 0\\ b & h & 1 \end{array}\right]$$*is the curve similarity transformation matrix, where the meanings of a, b, k, h are the same as those described in Formula (1), corresponding to the translation and scale transformation of the horizontal and translation directions, respectively*. ${F^{\prime }}_{A}=T\cdot F_{A}=\left\{\left(x_{1}^{*},\;y_{1}^{*}),(x_{2}^{*},\;y_{2}^{*}),\cdots ,(x_{m}^{*},\;y_{m}^{*}\right)\right\}$*is called the similarity transformation curve of F_A_, where:*(4)$$\left\{ \begin{array}{l} x_{i}^{*}=a\cdot x_{i}+b\\ y_{i}^{*}=k\cdot y_{i}+h \end{array}\right.$$

Without any doubt, the transformation matrix *T* can be more complicated, such as rotation, mirroring and shearing transformation, and various combinations thereof. Moreover, the definition of discrete curve similarity distance under the condition of curve similarity transformation can be given.

**Definition** **6.***Given a discrete curve F_A_ = {(x_1_, y_1_), (x_2_, y_2_) … (x_m_, y_m_)} as a reference curve, another curve F_B_ = {(x’_1_, y’_1_), (x’_2_, y’_2_) … (x’_n_, y’_n_)} is considered as a comparison curve. F’_A_ = T • F_A_ is transformed by a similarity transformation matrix T from F_A_. Among all similarity transformation matrices T, the minimum distance of all the matching distances between F’_A_ and F_B_ is as follows:*(5)$$\begin{array}{l} dis(t,a,b,k,h)=\frac{1}{m}\sum_{i=1,j=i+t}^{m}\left|T(x_{i},y_{i})-({x^{\prime }}_{j},{y^{\prime }}_{j})\right|\\ \;=\frac{1}{m}\sum_{i=1,j=i+t}^{m}\left|\left(x_{i}^{*},y_{i}^{*})-({x^{\prime }}_{j},{y^{\prime }}_{j}\right)\right|\\ \;=\frac{1}{m}\sum_{i=1,j=i+t}^{m}\left|\left(a\cdot x_{i}+b,k\cdot y_{i}+h)-({x^{\prime }}_{j},{y^{\prime }}_{j}\right)\right|\\ \;=\frac{1}{m}\sum_{i=1,j=i+t}^{m}\left|\left(a\cdot x_{i}+b-{x^{\prime }}_{j},k\cdot y_{i}+h-{y^{\prime }}_{j}\right)\right|\\ \;=\frac{1}{m}\sum_{i=1,j=i+t}^{m}\sqrt{{\left(a\cdot x_{i}+b-{x^{\prime }}_{j}\right)}^{2}+{\left(ky_{i}+h-{y^{\prime }}_{j}\right)}^{2}} \end{array}$$(6)$$D(F_{A},F_{B})=D({F^{\prime }}_{A},F_{B})=\min\left\{dis(t,a,b,k,h)\right\},$$*which is called the curve similarity distance of F_B_ to F_A_ under the transformation T. The meanings of a, b, k, h are the same as those described in Formula (1) and Figure 2, corresponding to the translation and scale transformation of the horizontal and translation directions, respectively. Meanwhile, t is the starting point position of the optimal matching of the reference curve on the comparison curve*.

Likewise, the definition of discrete curve similarity is as follows:

**Definition** **7.***Given a discrete reference curve F_A_ and a comparison curve F_B_, for a given threshold ε, if D(F_A,_ F_B_) ≤ ε, then the curve F_B_ is called similar to curve F_A_, and vice versa*.

Finally, calculating the similarity distance of the comparison curve to the reference curve is equivalent to calculating the optimal matching of the reference curve in the sense of the average distance on the comparison curve after the transformation. Therefore, performing a similarity measure between two curves requires two steps, one is to calculate the optimal matching, and the other is to perform threshold discrimination.

In the classic curve matching algorithm DTW, once a curve of two curves is translated or stretched, the distance measurement between them will be modified. In the curve similarity measure process, the curve similarity distance calculation depends on a transformation matrix. The characteristics of the transformation matrix can well prevent the deformation process of the translation, stretching and rotation of the curve. However, the transformation matrix is sometimes difficult to solve directly.

Accordingly, an evolutionary computation (EC) [32] algorithm has been introduced to obtain the transformation matrix and the optimal matching interval between two curves by minimizing the matching distance. The specific search algorithm is introduced later.

#### 2.1.4. Matching Calculation

It can be observed from the definition that the curve similarity distance is the optimal matching of the reference curve with the comparison curve under different similarity transformations. Therefore, the matching distance between two curves under different CSTs can be obtained by random search algorithm. EC is a very effective algorithm for intelligent random search, where the similarity transformation matrix *T* is the parameter space of random search.

The whole idea of EC is to generate *S* random populations *POP* = {*POP(*0*)*, *POP(*1*)*, …, *POP(S −* 1*)*}, each of which corresponds to a set of parameters of CST *T* and a fitness value *fitness = D(F_A_, F_B_)* from Formula (6), where the *fitness* is smaller, the population is better. At the same time in each iteration search, each individual will generate new descendants near itself, and the best individuals can be chosen to enter the next iteration search.

The above process is repeated until the iterative search reaches the maximum number of iterations, at which point the optimal individual parameters and fitness will be regarded as CST and CSD, respectively.

At the same time, the parameters (*t*, *a, b, k, h**)*** should meet certain constraint conditions and adopt real coding for the signature curve matching. The boundary condition of the following system is 0 ≤ *t* < 2*n*, 0.90 ≤ *a* ≤ 1.10, 0.90 ≤ *k* ≤ 1.10, −100 ≤ *b* < 100*,* and −100 ≤ *h* ≤ 100, where *a* and *k* are elastic scales in the horizontal and vertical directions, and *b* and *h* are translations.

The Algorithm 1 process is as follows:
**Algorithm 1.** Optimal matching calculation by EC*i* = 0**For***j* = 0: *S* − 1 **Do**//random generation of parameters ***t***, ***a, b, k, h***initial a population *POP_i_*(*j*) = (***t***, ***a, b, k, h***)calculate fitness of each member in *POP_i_*(*j*)**End For****While***i* < ***Iterations***
**Do** *i* = *i* + 1 //sorting and classification order *POP_i_*_−1_ by fitness in ascending and divide them into 4 levels //acceleration search **For**
*j* = 0:*S −* 1 **Do**  **If**
*POP_i_*_−1_(*j*) is at level *kind*   *//*where*, kind =* 1, 2, 3, 4, indicating the classification level of each population   //random generation of new parameters ***t***, ***a, b, k, h*** in the neighborhood   //i.e., ***a*** = rand(*POP_i−_*_1_(*j*).***a,***
*kind*)    generate new *kind*+1 subpopulations *nPOP* from *POP_i−_*_1_(*j*)   calculate fitness of each member in *nPOP*   //sorting subpopulations   order *nPOP* by fitness in ascending    //select best subpopulation *nPOP* as *POP_i−_*_1_(*j + S*)    *POP*_*i*−1_(*j* + *S*) = *nPOP(0)*  **End If** **End For** //global selection order *POP_i−_*_1_ by fitness in ascending, where there are 2*S* populations select the top *S* from *POP_i−_*_1_ as *POP_i_***End While**

When generating new *kind* + 1 subpopulations from one population, each subpopulation has different parameter generation range at different level. Generally speaking, the *fitness* is smaller, the parameter generation range of each subpopulation is narrower.

Suppose that the *j*-th population *POP*(*j*) is at the *kind*-th level, on which parameter generation range is *GR_kind_ = kind • GR_max_/4*, *kind +* 1 subpopulations *nPOP* can be randomly generated, here:(7)$$nPOP=\left\{nPOP(k)|\left|nPOP(k)-POP(j)\right|<\;GR_{kind},k=0\sim kind\right\}$$

For example, for parameter *a*, *GR_max_* = max(*a*) − min(*a*). When *kind* = 1 and a new random number *r* = −1 or 1 is generated, *a^new^* = *a^old^* + *r • GR_kind_ = a^old^* + *r • GR_max_*/4. At the same time, check whether *a^new^* satisfies the boundary condition.

The other parameters *t*, *b*, *k*, *h* all perform similar operations. Therefore, from the new subpopulation process generated by the parameters represented by each original population, the parameters of the generated subpopulations are determined according to the classification level of the original population. Although this parameter is also random, it varies randomly to the original population by its classification level.

### 2.2. Proposed System

Online signature verification system can be discriminated by the similarity distance of two signature curves, where one signature can be called as the reference signature and the other can be called as the comparison signature. For instance, two signature trajectories can be considered as plane curves which can be also divided into *X* curves and *Y* curves, as shown in Figure 3.

For a reference signature curve and a comparison signature curve, if the similarity distance between them is sufficiently small, it can be considered that the comparison signature is genuine one, otherwise it is a forged signature. A block diagram of the proposed system is illustrated in Figure 4.

Given *SignR* = {(*x*_1_, *y*_1_), (*x*_2_, *y*_2_) … (*x_M_*, *y_M_*)} and *SignC* = {(*x’*_1_, *y’*_1_), (*x’*_2_, *y’*_2_) … (*x’_N_*, *y’_N_*)} as the reference curve and the comparison curve, respectively. For the calculation of the similarity distance of the signature curves, if the reference curve is calculated as a whole with the comparison signature, it will lead to greater errors. Segmentation curve matching can be used to better measure local differences between curves, which could be common in complex curve similarity measures.

#### 2.2.1. Preprocessing

As the variations in different signatures have different dynamic ranges, min–max normalization is implemented in their *X* and *Y* coordinates. One signature should be reprocessed as follows:(8)$$x_{i}=2000\times \frac{x-\min(x)}{\max(x)-\min(x)},y_{i}=1000\times \frac{y-\min(y)}{\max(y)-\min(y)}$$
where *x* and *y* are the original coordinates, and *x_i_* and *y_i_* are the normalized coordinates. The normalization scales of horizontal and vertical directions are different, and 2000 and 1000 are taken respectively to keep the original scale of the signature curve as much as possible.

#### 2.2.2. Segmentation

*SignR* is divided into *K* sections and there are *m* data points of each the reference segmentation curve. Each the reference segmentation curve can be defined as
(9)$$\left\{ \begin{array}{l} {\left(SignR\right)}_{i}=\{(x_{t},y_{t})|t\in [R_{i},R_{i}{}^{\prime }],R_{i}{}^{\prime }-R_{i}=m\}\\ m=\mathrm{INT}\left(M/K\right)\\ R_{0}=(M-m\cdot K)/2\\ R_{i}=R_{0}+(i-1)\cdot m\\ R_{i}{}^{\prime }=R_{0}+i\cdot m\\ i\in [1,K] \end{array}\right.$$
where, INT(*x*) is the integral function.

Likewise for the comparison curve, each possible matching interval corresponding to the reference segmentation curve can be defined as
(10)$$\left\{ \begin{array}{l} {\left(SignC\right)}_{i}=\{({x^{\prime }}_{t},{y^{\prime }}_{t})|t\in [C_{i},{C^{\prime }}_{i}],{C^{\prime }}_{i}-C_{i}=2n+m\}\\ n=\mathrm{INT}\left(N/K\right)\\ C_{0}=(N-n\cdot K)/2\\ C_{i}=C_{0}+(i-1)\cdot n-n/2-m/2\\ {C^{\prime }}_{i}=C_{0}+(i+1)\cdot n-n/2+m/2\\ 0\leq C_{i}<{C^{\prime }}_{i}\leq N,i\in [1,K] \end{array}\right.$$

Here, considering the positional correlation and the deviation between the signature curves, when the reference signature length of each segmentation is *m*, the interval to be matched of the comparison signature is at least *m* in length, and is offset by *n* before and after the corresponding segmentation position. Equivalently, the reference curve swims within the interval to be matched on the comparison curve to obtain the optimal matching position.

#### 2.2.3. Segmentation Matching

The process of the optimal segmentation matching is as follows:

*Step 1*: Take the reference signature curve as a template, and divide it into *K* segments according to Equation (9).

*Step 2*: The comparison curve should be divided into *K* segmentations according to Equation (10).

*Step 3*: For the *i*-th segmentation of the comparison curve, search the optimal matching with the corresponding the *i*-th segmentation of the reference curve by EC algorithm, and get the similarity distance *d_i__._* Meanwhile, the matching distance *dx_i_* and *dy_i_* of the corresponding *X*, *Y* curves can be calculated based on the current matching result.

*Step 4*: Set *σx_i_*, *σy_i_* which are the standard deviation of *X*, *Y* data points in *i*-th segmentation of the reference curve, compare it with the matching distance *dx_i_* and *dy_i_*, and calculate the similarity score *sx_i_* and *sy_i_* of this segmentation, respectively, as in Equations (11) and (12).
(11)$$\left\{ \begin{array}{l} sx_{i}=h(dx_{i},\sigma x_{i},\alpha )+h(dx_{i},\sigma x_{i},\beta )+h(dx_{i},\sigma x_{i},\gamma )\\ sy_{i}=h(dy_{i},\sigma y_{i},\alpha )+h(dy_{i},\sigma y_{i},\beta )+h(dy_{i},\sigma y_{i},\gamma ) \end{array}\right.$$
(12)$$\left\{ \begin{array}{l} h(d,\sigma ,\delta )=100\cdot \exp(-0.5\delta \cdot d^{2}/(\sigma /4+10{)}^{2})\\ \alpha =0.25\sim 0.5,\beta =1\sim 2,\gamma =2\sim 5 \end{array}\right.$$

*Step 5*: Repeat step 3 until all segmentation curves parameters are calculated.

*Step 6*: Calculate the average of *sx*, *sy* as the result outputs of *X*, *Y* curves’ similarity measure, and use the weighted average as the result output of the similarity measure *Score* of the two curves, as shown in later Formula (14).

It should be noted that when performing the optimal matching segmentation calculation, the similarity distance between the two curves can be obtained, which are considered as 2D curves. Next, the matching distances of the corresponding 1D curves *X* and *Y* can be separately calculated. Obviously, the calculated matching distances are absolute values, and if a similarity evaluation is to be performed, one threshold is needed to discriminate. For this reason, the similarity average is calculated using Gaussian functions of three different widths and the absolute distance measure is converted to a relative measure between 0 and 100. Thus, the discrimination threshold can be unified to a value between 0 and 100.

A pair of genuine signature curves is adopted for the optimal segmentation matching as seen in Figure 5. The matching results of *X*, *Y* curves are shown in Figure 6, respectively.

A pair of genuine and forged signature curves is adopted for the segmentation matching as seen in Figure 7. The matching results of *X*, *Y* curves are shown in Figure 8, respectively.

The optimal segmentation matching results between three curves are shown in Table 1 and Table 2.

It can be seen from Table 1 and Table 2 that for the similarity of *X*, *Y* curves is calculated by the same template signature and segmentations, the similarity between two genuine signatures is usually higher than that between the genuine and forged signatures. On each segmentation, the similarity calculation of the *X*, *Y* curves depends on the standard deviations of the respective segmentations in the template signature, as shown in Equation (11). To accurately estimate this deviation, a large number of genuine and forged signatures are needed for matching calculations and statistics. Obviously, this is difficult to obtain in practical applications. Here we use the intra-segmentation standard deviation of the signature segmentation itself and a deviation as the empirical value. In addition, the three control parameters *α*, *β*, and *γ* are equivalent to controlling the width of the Gaussian function, and are also an empirical value. Here, *α* = 0.4, *β* = 1.6, *γ* = 3.2 are selected. The changes in these parameters are not sensitive to the correct rate of the final evaluation results. Due to space limitations, this article will not discuss them.

#### 2.2.4. Feature Extraction

In the *i*-th matching interval [*R_i_*, *R_i_*_+1_], the interval velocity ratio (*IVR*) of corresponding points is calculated as follows:(13)$$\left\{ \begin{array}{l} v_{j}=\sqrt{{\left(x_{j}-x_{j-1}\right)}^{2}+{\left(y_{j}-y_{j-1}\right)}^{2}}\\ {v^{\prime }}_{j}=\sqrt{{\left({x^{\prime }}_{j}-{x^{\prime }}_{j-1}\right)}^{2}+{\left({y^{\prime }}_{j}-{y^{\prime }}_{j-1}\right)}^{2}}\\ IVR_{i}=\frac{100}{{R_{i}}^{\prime }-R_{i}+1}\sum_{j=R_{i}+1}^{{R_{i}}^{\prime }}\min(\frac{v_{j}+1}{{v^{\prime }}_{j}+1},\frac{{v^{\prime }}_{j}+1}{v_{j}+1}) \end{array}\right.$$

#### 2.2.5. Similarity Measure

The similarity measure *Score* of two signature curves is calculated as follows:(14)$$\left\{ \begin{array}{l} Score=w_{a}\cdot LSC+w_{b}\cdot GSC\\ LSC=\frac{1}{K}\sum_{i=1}^{K}\left(0.2sx_{i}+0.3sy_{i}+0.5IVR_{i}\right)\\ GSC=g(M/N)\\ w_{a}+w_{b}=1,w_{a}\geq 0,w_{b}\geq 0 \end{array}\right.$$
where *LSC* and *GSC* are local similarity score and global similarity score, respectively, while *w_a_* and *w_b_* are the corresponding weights. *M* and *N* are the lengths of the reference signature and the comparison signature, respectively.

Here:(15)$$g(x)=\{\begin{array}{cc} 0 & x<0.5\\ 100\times \exp(-2{\left(x-1\right)}^{2}) & 0.5\leq x\leq 2\\ \begin{array}{cc} 0 & \end{array} & x>2 \end{array}$$
is used to calculate score of the writing time ratio of two origin signatures.

The calculation of the weight is calculated by enumeration, where *w_a_* = 0.85 and *w_b_* = 0.15, and the detailed process is shown in the next Section 3.4.

It is considered that threshold *ε* of the signature verification system is 60 in many cases, and when *Score* is greater than 60, it may be distinguished into a genuine signature, and below 60 may be considered as a forged signature.

This is similar to the 100 point test. Passing more than 60 points is a pass, and below 60 is a failure. Of course, accurately determining the threshold is also a problem that needs to be studied in depth. For each user’s signature threshold determination, some other real registration signatures or even skilled forged signatures are needed. As a single template signature authentication system, only a reasonable threshold is given here, which is one of the key issues that need to be studied in the future.

## 3. Experiments

In this section, experiments to evaluate the efficacy of the four datasets are described and signature verification performances are reported.

### 3.1. Dataset and Evaluation Protocol

The efficacy of the proposal is demonstrated on the publicly available SUSIG, SVC2004 Task1&Task2 and MCYT datasets. The main differences among the four datasets are the acquisition protocol, device, and signer. In the following, the datasets used in this paper are briefly described:(1)SUSIG Visual Subcorpus [34]: This dataset consists of 2820 western signatures from 94 signers with 20 genuine signatures collected in two sessions and 10 skilled forgery signatures (half are highly skilled) with an LCD touch device. For convenience, this subcorpus is called SUSIG for short in this paper. The data in SUSIG consists of *X*, *Y*, pressure, and timestamp, collected at 100 Hz.(2)SVC2004 Task1 Subcorpus [35]: This datasets are acquired with a Wacom graphic tablet. It consists of 800 English and Chinese signatures from 40 signers with 20 genuine signatures collected in two sessions and 20 skilled forgeries per signer. For convenience, this subcorpus is called SVC1 for short in this paper. The data in SVC1 consists of *X*, *Y* and timestamp, collected at 100 Hz.(3)SVC2004 Task2 Subcorpus [35]: It is also composed of 40 signers with the same number of genuine and forged signatures as in Task1. For convenience, this subcorpus is called SVC2 for short in this paper. The data in SVC2 consists of *X, Y*, pressure, azimuth, altitude, timestamp, and button status, collected at 100 Hz.(4)MCYT-100 Subcorpus [36]*:* It is also composed of 100 signers with 25 genuine and 25 forged signatures. For convenience, this subcorpus is called MCYT for short in this paper. The data in MCYT consists of *X*, *Y*, pressure, azimuth, altitude, timestamp, and button status, collected at 100 Hz.

Out of these, one genuine signature is selected randomly to be used as reference sample (template), and the other genuine signatures and all skilled forgeries are used as test samples. Thus, in our work there are 9480 reference and test signatures from 274 signers to be verified in total.

We adopt EER, i.e., the error rate at which false acceptance rate (FAR) and false rejection rate (FRR) are equal, as a measure for characterizing verification performance. In order to obtain reliable results for independent test data, this process of random selection of reference signatures and performance evaluation is repeated ten repetitions.

### 3.2. Parameter Determination for EC

Taking the template signature of Figure 5 as an example, the signature is from SVC2004 Task1 signer #1, with 147 data points. The total number of segmentations *K* of different data points of the reference signature is shown in Table 3.

The optimal segmentation matching calculation between the template signature and itself was executed. The theoretically similarity distance of each segmentation is theoretically zero. The parameters *S* and *Iterations* of the EC are determined by enumeration calculation as seen in Figure 9, where *S* = 4, 8, 12, 16, 20 and *Iterations* = 100, 200, 400, 800. In the abovementioned optimal segmentation matching calculation, although the global optimal parameters are not obtained, the different *S* and *Iterations* can quickly reach the local minimum. Comprehensive calculation of speed and accuracy requirements, choose *S* = 20 and *Iterations* = 400 as the control parameters for EC.

### 3.3. Feature Validity Test

Select 20 genuine signatures and 20 skilled forged signatures of the first signer on SVC1, take one of the first 10 genuine signatures as a template in turn, and the remaining 19 genuine signatures and 20 skilled forged signatures respectively perform matching calculation, and then calculate *sx*, *sy*, *IVR*, *GSC*, *LSC*, and final *Score*. 

Figure 10a–d show the distribution of similarity differences between genuine and skilled forged signatures. Signature verification can be regarded as a two-category problem. It can be seen from the distribution process of the comparison results of the signatures in Figure 10a that only the similarity of *X* and *Y* curves are used, and it is difficult to distinguish the authenticity of each test signature. From Figure 10b, it can be seen that the IVR has a certain degree of discrimination, but there are many indistinguishable confusing signatures. However it can be clearly seen in Figure 10c that the fusion feature LSC and the global feature GSC have a high degree of discrimination. Only a very small number of test signatures are misidentified. In the Formula (11), the above several features are merged, and the similarity of the two curves can be used as a one-dimensional index. The discriminant threshold can be used to directly identify or classify the signature authenticity.

In Figure 10d, it can also be clearly seen that different template signatures have different discriminating thresholds, and the degree of discrimination between genuine and forged signatures is also different. Using the #4, #5, and #7 signatures as templates, the signature authenticity can be completely distinguished, and the #7 template has the largest degree of discrimination.

### 3.4. Feature Weight Calculation

Let the number of signatures of the template equal one and the weight value *w_a_* is increments from 0 to 1 with the interval 0.05. The signatures of the first four signers on SVC1 dataset are selected for training to find the optimal *w_a_*. On the four signature datasets, the respective EERs under different weights are calculated. The results are shown in Table 4.

From Table 4, it can be seen on the training samples that the minimum EER is obtained when *w_a_* = 0.85, and the EERs on the other datasets are 3.47%, 12.30%, 12.25%, and 6.07%, respectively. At this point, the same minimums are obtained on SVC1 and MCYT datasets. In Figure 11, we can see that when *w_a_* is incremented, the EER value changes to a convex function, and the EER on different datasets has only a minimum value.

In fact, we can also see that the EER is not much different when *w_a_* ∈ [0.6, 0.95], which means that our weight selection has better robustness, and when *w_a_* ≤ 0.3, The EER has increased dramatically. At the same time, when *w_a_* = 0, signature verification actually only depends on GSC the signature writing time ratio. On the SVC1 and SVC2 datasets, the EER of each is over 20%, and on the SUSIG dataset, the EER is still less than 5%. It can be seen that the writing time ratio is a better feature distinguishing the test signature. In addition, SVC1&SVC2 is much higher than the SUSIG dataset in the difficulty of signature verification of four signature datasets. This can also be obtained from the whole experimental results.

### 3.5. Experimental Results

Performance of the system with the maximum EER, the minimum EER, the average EER and the standard deviation of EERs measured in percentage for different number reference signatures of similarity metrics are shown in Table 5. 

From the results described above, when experiments are implemented on SUSIG, EER = 3.47% can be the best result based on CSM with five reference samples. As for SVC1&SVC2&MCYT, it can be provided EER = 12.30%, EER = 12.25% and EER = 6.07%, respectively. For four different datasets and different number of reference samples, the EERs of test results with #1 template repeated 10 times are arranged in ascending order, as shown in Figure 12.

At the same time, it should be emphasized that the deviation of the maximum and minimum values of EER is more than double almost when #1 sample is randomly selected as templates in ten repetitions as seen in Table 5 and Figure 12. It demonstrates that the selection of template samples is also essential and representative template samples can effectively improve the accuracy of signature verification.

In order to demonstrate the effectiveness of our proposed method, we compare the results of our proposed method with other state-of-the-art methods. It is to be mentioned that each of these methods have different features and classifiers, and it is difficult to make comparisons between them based on different datasets. Hence, we just compare the performance of methods which are carried out on SUSIG, SVC1, SVC2 and MCYT datasets. Nevertheless, the best results methods carried out both on them are taken for comparative studies, which use five genuine reference signature of a signer for enrollment. The best EERs reported from the reference works on SUSIG, SVC1, SVC2 and MCYT are given in Table 6, Table 7, Table 8 and Table 9 with one and five genuine reference signatures.

Furthermore, it should be pointed out that the performance of a signature verification system is related to the number of samples used to build the model. Even in some statistical models, true and false signatures need to be trained. However, in many cases, it is difficult to register a large number of signatures in the actual system, which also limits the practical applications of many excellent methods, but for our signature verification system, the requirement for the number of reference signatures or templates is minimal and even only one signature can be used. Moreover, among the already known single signature systems, our performance is the best, and is very close to that of multi-signature systems.

## 4. Conclusions

The similarity measurement of curves is an old problem. A lot of pattern recognition problems can be converted into curve similarity problems to study. In this research, we presented a novel signature verification based on the curve similarity model, which is equally competitive when compared to other approaches and leads to much simpler and easier matching procedures. Considering internal and external writing environments being always varied, signatures were effectively aligned to the reference signature curve by CSM and a curve similarity distance was proposed to make an assessment the similarity between test signatures and references. Open access signature datasets SUSIG, SVC2004 Task1&Task2, and MCYT-100 were used in our work, and several experiments were implemented. Experimental results illustrated that the best matching could be obtained by our proposed CSM method with one signature template. The error rates EER_SUSIG_ = 3.47%, EER_SVC1_ = 12.30%, EER_SVC2_ = 12.25% and EER_MCYT_ = 6.07% were provided, respectively, which demonstrated the effectiveness and robustness of our proposed method. The most important thing is the case that our method can use one signature to authenticate, and the performance of our method is not much different from that of multi-signature verification systems. Finally, this innovative method opens the door to new competitions on signature verification using a single signature as reference template.

## Figures and Tables

**Figure 1 sensors-19-04858-f001:**
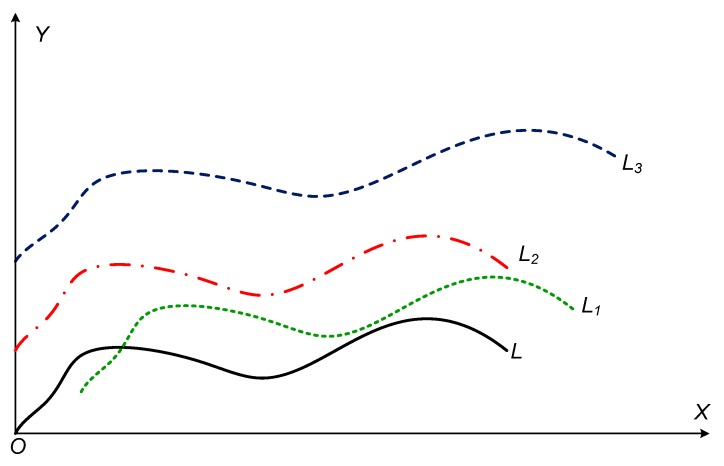
Translation and stretching transformation of curves.

**Figure 2 sensors-19-04858-f002:**
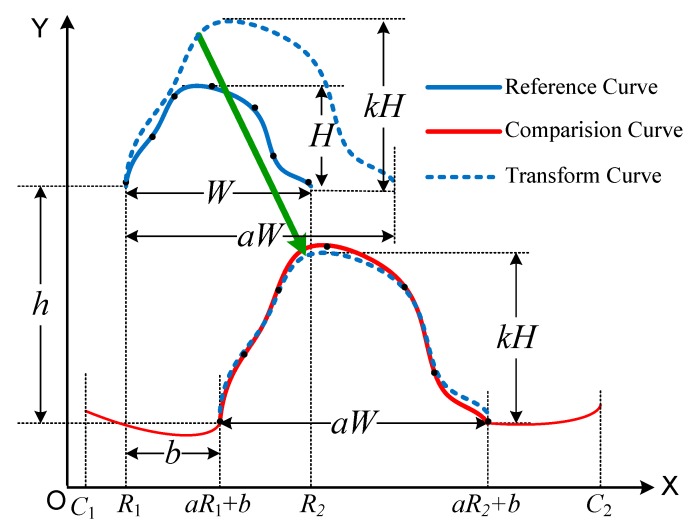
The curve similarity transformation between the two curves.

**Figure 3 sensors-19-04858-f003:**
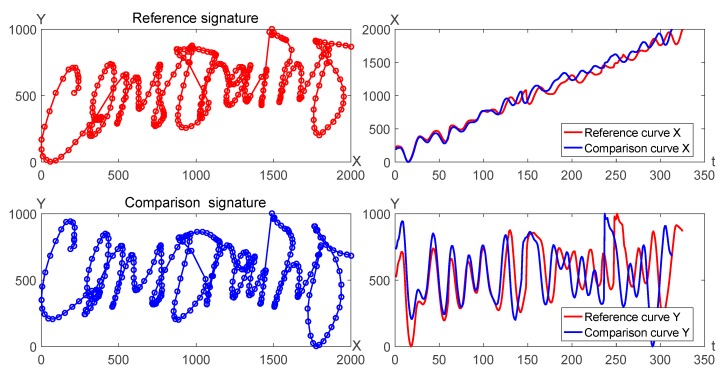
Two signature curves and their corresponding *X* and *Y* curves.

**Figure 4 sensors-19-04858-f004:**
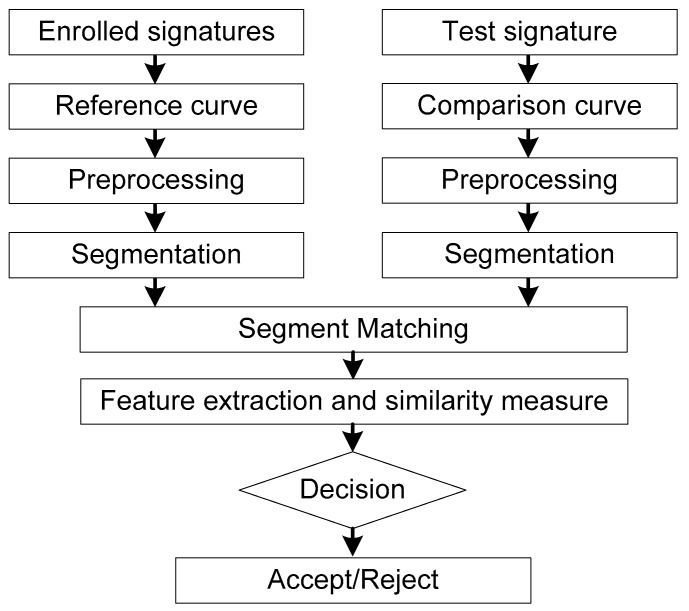
Block diagram of the proposed system.

**Figure 5 sensors-19-04858-f005:**
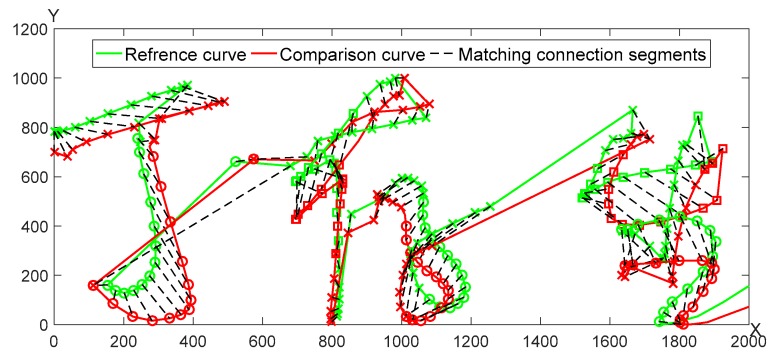
The matching results of two genuine signatures.

**Figure 6 sensors-19-04858-f006:**
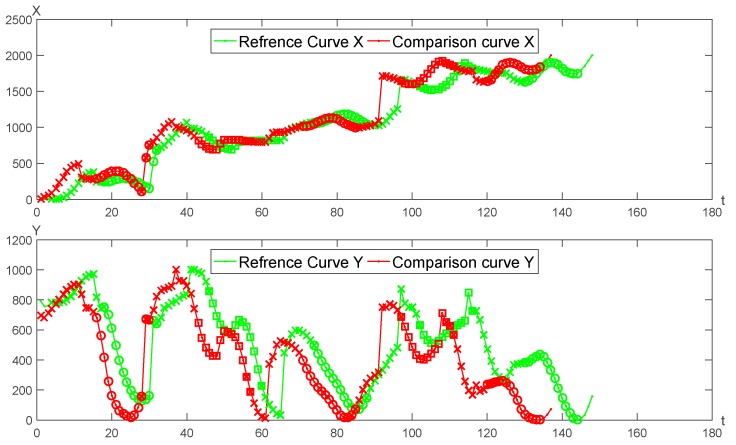
The matching result of *X* and *Y* curves between two genuine signatures.

**Figure 7 sensors-19-04858-f007:**
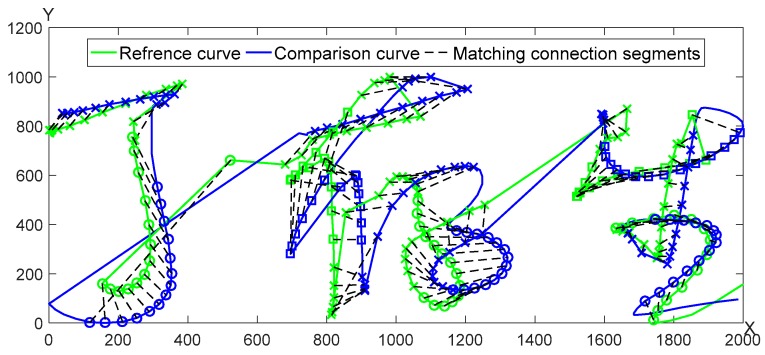
The matching results of the genuine and the forged signatures.

**Figure 8 sensors-19-04858-f008:**
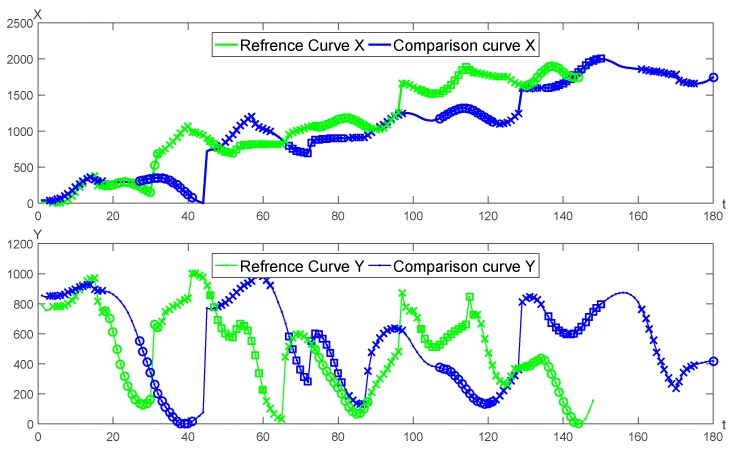
The matching result of *X*, *Y* curve between the genuine and forged signatures.

**Figure 9 sensors-19-04858-f009:**
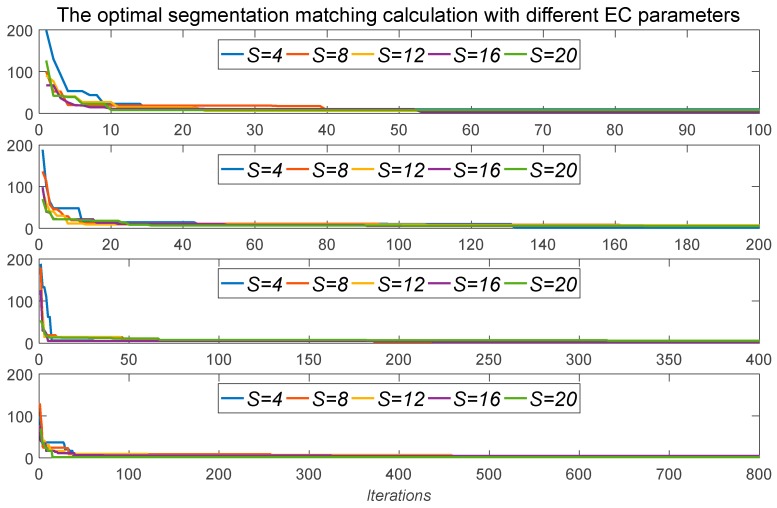
The optimal segmentation matching calculation with different EC parameters.

**Figure 10 sensors-19-04858-f010:**
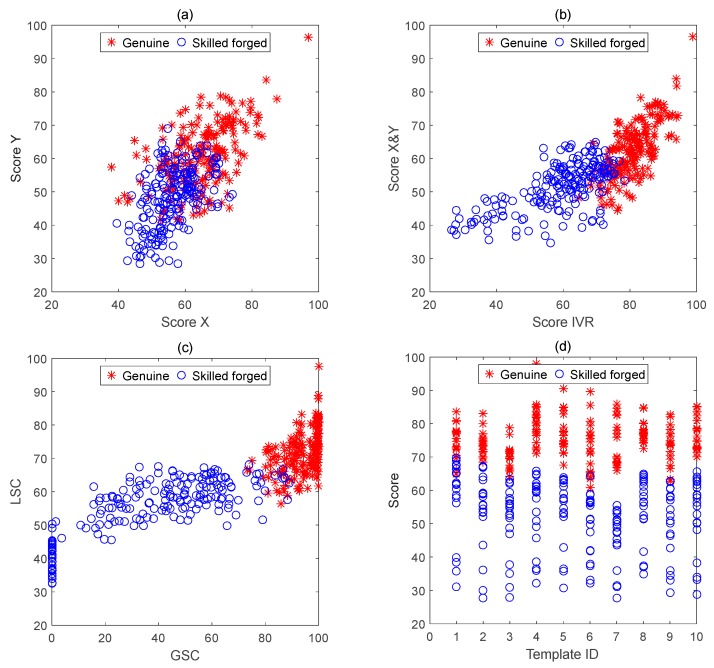
One of the first 10 genuine signatures is used as the template signature in turn, and the remaining genuine and skilled forged signatures are used as the comparison result of the test signatures. (**a**) Score *X* and score *Y* distribution; (**b**) Score *IVR* distribution; (**c**) *GSC* and *LSC* distribution; (**d**) 10 templates in turn *Score* distribution.

**Figure 11 sensors-19-04858-f011:**
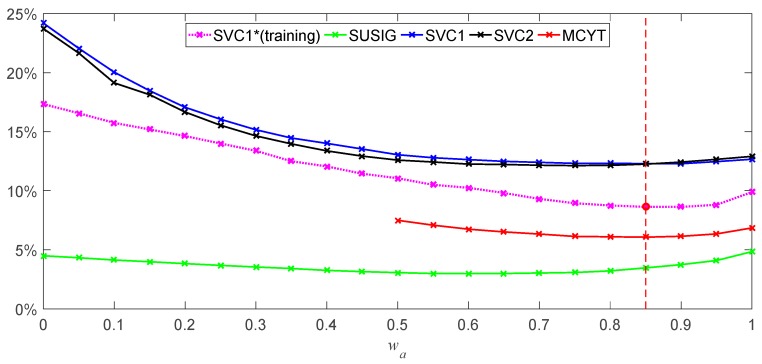
EERs and the mean on four signature datasets with different weight *w_a_*.

**Figure 12 sensors-19-04858-f012:**
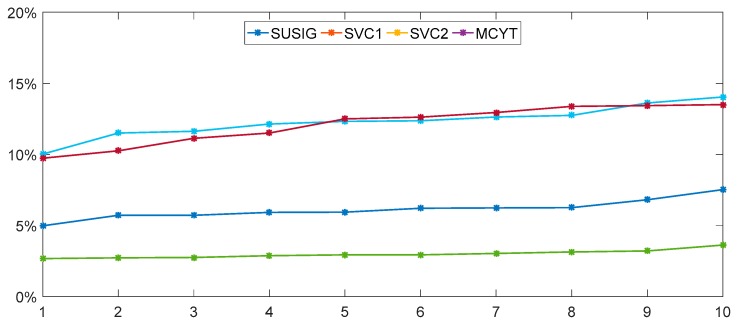
EERs of test results with #1 template repeated 10 times for SUSIG, SVC1, SVC2 and MCYT, respectively.

**Table 1 sensors-19-04858-t001:** Results of segmentation matching between a pair of genuine signature curves (*K* = 10).

No.	[*R_i_, R_i_’*]	Matching	*t*	*a*	*b*	*k*	*h*	*d*	*dx*	*dy*	*ox*	*oy*	*sx*	*sy*
1	[3,17]	[0,14]	0	1.03	36	0.93	10	44.6	22.4	27.8	70.5	92.3	77.9	69.2
2	[17,31]	[15,29]	15	0.99	9	0.92	95	61.6	11.7	42.4	79.7	220.1	94.1	81.6
3	[31,45]	[27,41]	25	1.05	−23	0.90	29	55.4	20.7	34.3	54.7	107.9	73.8	65.8
4	[45,59]	[42,56]	27	0.99	−11	0.90	61	55.4	13.9	47.9	23.5	130.2	67.8	57.8
5	[59,73]	[55,69]	27	0.90	6	0.90	61	26.3	3.9	21.8	66.9	234.3	99.1	95.0
6	[73,87]	[70,84]	29	1.06	−86	0.90	68	18.5	6.6	8.6	20.9	112.1	89.5	97.1
7	[87,101]	[82,96]	28	1.06	−61	1.09	17	66.3	24.7	40.4	151.5	251.8	91.3	86.3
8	[101,115]	[96,110]	29	1.02	31	0.91	62	66.3	16.6	47.4	77.2	108.4	88.2	50.9
9	[115,129]	[108,122]	28	0.99	8	1.01	37	32.6	24.8	14.7	35.3	137.7	52.2	94.2
10	[129,143]	[119,133]	26	0.92	66	0.92	100	44.7	11.0	43.8	42.5	154.2	87.6	68.6
Average													82.1	76.7

**Table 2 sensors-19-04858-t002:** Results of segmentation matching between a pair of genuine and forged signature curves (*K* = 10).

No.	[*R_i_, R_i_’*]	Matching	*t*	*a*	*b*	*k*	*h*	*d*	*dx*	*dy*	*ox*	*oy*	*sx*	*sy*
1	[3,17]	[2,16]	2	0.96	32	0.94	−95	56.4	15.1	30.2	70.5	92.3	88.6	65.6
2	[17,31]	[26,40]	26	0.91	53	0.9	94	112.0	96.7	172.2	79.7	220.1	20.9	23.6
3	[31,45]	[47,61]	31	1.04	30	0.93	−63	38.3	9.0	16.6	54.7	107.9	93.8	89.2
4	[45,59]	[66,80]	30	1.07	13	0.91	53	97.0	26.6	76.9	23.5	130.2	38.3	36.4
5	[59,73]	[82,96]	26	0.94	82	0.9	−40	50.2	10.7	59.3	66.9	234.3	93.5	71.8
6	[73,87]	[106,120]	30	1.00	82	0.9	−12	28.6	7.9	18.7	20.9	112.1	85.3	87.4
7	[87,101]	[119,133]	23	0.90	76	1.09	−20	61.7	16.4	39.9	151.5	251.8	96.0	86.6
8	[101,115]	[135,149]	19	0.95	96	0.91	−30	37.1	12.3	29.1	77.2	108.4	93.2	72.9
9	[115,129]	[160,174]	24	0.95	40	0.9	41	19.6	4.5	10.4	35.3	137.7	97.1	97.0
10	[129,143]	[179,193]	23	1.00	3	0.9	24	39.6	17.3	16.6	42.5	154.2	73.9	93.9
Average													78.1	72.4

**Table 3 sensors-19-04858-t003:** Proposal parameter *K* with different data points of the reference signature.

Order	1	2	3	4	5	6	7	8	9	10
Data Points Count *M*	≤100	≤150	≤200	≤250	≤300	≤350	≤400	≤500	≤600	Other
Proposal *K*	8	10	12	14	16	18	20	22	24	30

**Table 4 sensors-19-04858-t004:** EERs under different weights on four datasets.

#	Weight		Average EER (in %)				
			Train	Test			
	*w_a_*	*w_b_*	SVC1 *	SUSIG	SVC1	SVC2	MCYT *
1	0.00	1.00	17.34	4.49	24.20	23.73	
2	0.05	0.95	16.56	4.32	22.06	21.66	
3	0.10	0.90	15.75	4.13	20.04	19.13	
4	0.15	0.85	15.19	3.98	18.47	18.13	
5	0.20	0.80	14.63	3.83	17.07	16.66	
6	0.25	0.75	14.00	3.68	16.05	15.54	
7	0.30	0.70	13.38	3.54	15.16	14.65	
8	0.35	0.65	12.50	3.41	14.46	13.97	
9	0.40	0.60	12.06	3.26	14.01	13.38	
10	0.45	0.55	11.44	3.15	13.53	12.93	
11	0.50	0.50	11.06	3.06	13.05	12.59	7. 50
12	0.55	0.45	10.50	2.99	12.79	12.43	7.08
13	0.60	0.40	10.25	2.98	12.64	12.25	6.74
14	0.65	0.35	9.81	2.99	12.49	12.22	6.51
15	0.70	0.30	9.31	3.03	12.40	12.16	6.33
16	0.75	0.25	8.94	3.08	12.32	12.13	6.13
17	0.80	0.20	8.75	3.21	12.32	12.15	6.10
18	0.85	0.15	8.63	3.47	12.30	12.25	6.07
19	0.90	0.10	8.64	3.74	12.32	12.42	6.14
20	0.95	0.05	8.81	4.09	12.47	12.66	6.34
21	1.00	0.00	9.88	4.84	12.67	12.91	6.85

* Only choose *w_a_* = 0.5~1.0 as there is a long computation time with the most signatures.

**Table 5 sensors-19-04858-t005:** Performances of the system with different dataset.

DataSet	# of Samples	EER (in %)		
		Average	Minimum	Maximum
SUSIG	1	3.47	2.27%	4.32%
SVC1	1	12.30	9.62%	14.94%
SVC2	1	12.25	9.53%	14.58%
MCYT	1	6.07	3.98%	7.92%

**Table 6 sensors-19-04858-t006:** Comparative studies of state-of-the-art methods implemented on SUSIG.

Method	# of Samples	Average EER (in %)
Fuzzy modeling [37]	5	5.38
Histogram + Manhattan [38]	5	4.37
FFT + DTW [25]	5	3.03
DTW_Linear C [34]	5	2.10
35 global feature + FLD [39]	5	1.59
Parzen window + DCT [23]	5	1.49
TASS + RLCSS [26]	5	0.52
Target-wise [28]	1	6.67
Proposed method	1	3.47

**Table 7 sensors-19-04858-t007:** Comparative studies of state-of-the-art methods implemented on SVC1.

Method	# of Samples	Average EER (in %)
DTW [40]	5	6.96
DTW + HMM [18]	5	6.91
LCSS + SVM [22]	5	6.84
Wavelet Packet [41]	5	6.65
TASS + RLCSS [26]	5	5.33
SVC-competition [35]	5	2.84
Target-wise [28]	1	17.25
Proposed method	1	12.30

**Table 8 sensors-19-04858-t008:** Comparative studies of state-of-the-art methods implemented on SVC2.

Method	# of Samples	Average EER (in %)
Fuzzy modeling [37]	5	7.57
Function-based + HMM [42]	5	7.14
LCCS-SVM [22]	5	6.84
DTW + HMM [18]	5	6.91
LCSS [26]	5	5.33
Feature selection + DTW [43]	5	3.38
SVC-competition [35]	5	2.89
5 features DTW + VQ [15]	5	2.73
DTW with SCC [17]	5	2.63
Target-wise [28]	1	18.25
Proposed method	1	12.25

**Table 9 sensors-19-04858-t009:** Comparative studies of state-of-the-art methods implemented on MCYT.

Method	# of Samples	Average EER (in %)
DTW + Fourier descriptors [25]	5	7.22
Symbolic Representation [4]	5	6.12
HMM+Parzen Window [44]	5	5.29
Time function_LDP [45]	5	5.20
Histogram + Manhattan [38]	5	4.02
Neuro-fuzzy system [46]	5	4.02
Fusion matchers [47]	5	3.81
Dynamic programming [48]	5	3.52
HMM + Viterbi Path [49]	5	3.37
Wavelet coefficients [28]	5	3.21
GMM + DTW [16] Velocity and pressure partition [50]	5	3.05
	5	1.09
Target-wise [28]	1	13.56
Proposed method	1	6.07

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
