# Peer review of "Online Signature Verification Based on a Single Template via Elastic Curve Matching"

_sensors, 2019, doi:10.3390/s19224858_

Round 1

Reviewer 1 Report

In the matching process of signature curves, the research design a sectional optimal matching algorithm. They develop a new consistent and discriminative fusion feature extraction for identifying the similarity of signature curves. The experimental results show that their system achieves the same performance with five samples assessed with multiple state-of-the-art automatic signature verifiers and multiple datasets. The paper is interesting and well written and fits with the topics of the Journal. Some minor concerns are:

1.line 316 formula (11) should be formula (7)

2.line 317 formula (12) should be formula (8)

3. line 329 formula (15) should be formula (9)

4. line 325 Table 2, the last line should be on the same page with other data.

5. line 439 Table 5, the last line "MCYT" should be on the same page with other data.

6. line 475 Table 9 , the title should be on one line, and because the data over one page, the title should be cross page.

Author Response

Thank you for your evaluation and careful review of the paper, and we have made changes to the corresponding minor issues. In addition, we updated some of the diagrams and added Section 3.4 to illustrate the validity of feature extraction. There are some minor bugs that we have fixed.

Reviewer 2 Report

The paper present a new elastic curve matching method for online signatures that exploits just one reference. Experimental results have been demonstrate with respect to publicly available databases, SUSIG, SVC 2004 and MCYT.

In general terms, the paper is interesting but many aspects need to be improved. Overall, the presentation of the content and the organization of the paper must be improved.

In my opinion, two are the main lacks of the paper:

A) The paper is not well organized. There is a mixture between the presentation of the method and the experimental procedure. Many things seem the results of arbitrary choices made by the authors.

B) The presented system requires to set an huge amount of parameters and thresholds. The authors have to split each dataset in training and test set. They have to use all the training sets or just one for computing the values of all parameters and thresholds. Eventually, they have too verify the performance of the system only on the test set. It seems that the authors computed the parameters and evaluated the system on the same set of data. This choice doesn't allow to verify the goodness of the method and doesn't allow to compare the proposed systems with the others in the state of the art.

My suggestions:

1) The sentence "By consistent and discriminative ..." starting at line 44 is not clear. Write it again.

2) Among the methods based upon DTW, you didn't cite the most recent paper on signature verification, which has been tested in MCYT dataset. I suggest you to read and cite the following paper: Parziale, A., Diaz, M., Ferrer, M. A., & Marcelli, A. (2019). SM-DTW: Stability Modulated Dynamic Time Warping for signature verification. Pattern Recognition Letters, 121, 113-122.

3) Among the methods that exploit the concept of LCSS, I suggest you to read and cite the following paper, which has been tested on SVC datatset: Parziale, A., Marcelli, A., Exploiting stability regions for online signature verification (2014) Advances in Digital Handwritten Signature Processing: A Human Artefact for E-Society, pp. 13-25.

4) In the introduction, I suggest you to introduce some novel methods that try to transform an offline signature in an online signature. Only for example, a method also based upon DTW is: Diaz, M., Ferrer, M.A., Parziale, A., Marcelli, A.; Recovering Western On-Line Signatures from Image-Based Specimens (2018) Proceedings of the International Conference on Document Analysis and Recognition, ICDAR, 1, pp. 1204-1209.

5) Line 170: A symbol is missed in the formula. 

6) dis(a.b.k.h) has been used in formula 1 but has never introduced before

7) Line 199 and 201: Missing symbols in the formula

8) Line 235: If populations are S, populations have to be indexed as POP (0) to POP(S-1) and not as POP(1)..POP(S-1) 

9) Line 244: It is not clear how the boundary constrains for t, a, k,b,h has been defined. Are these limits indipendent from the dataset?

10) Line 258: I'm sure that the algorithm is not correct. Instead of the "IF POP at level kind" you have to use a "For each kind"

11) Line 260: t= rand(Pop.t, kind) has to be introduced. You have to explain why rand is a function of both t and kind

12) Line 273 - 280: These sentences and formula are not clear. Write it again. I'm not able to understand the link between the fitness and the parameter generation range.

13) Formula (6): Why do the scale factor is different between x and y? 

14) Formula (7) and (8): There are missing symbols. It is not clear Ci'-Ci is equal to 2n+m. Both the signatures are divided in k parts. Why do the compared signature dipend on the number of point of the reference?

15) Line 311: The sentence has no sense. It seems that a part is missed.

16) Line 320: If I understood well, in Step 3 you wrote that EC is applied 3 times: once for computing di, once for dix and eventually for diy. Is it useful to compute di? 

17) Line 323 Step 4: What is the rational of comparing the std of the points with the distances computed by EC?

18) Line 329: Formula 15 is referenced but it doesn't exist

19) Figure 5: Please, introduce a label for the blue line

20) Figure 7: Please, introduce a label for the red line

21) Line 343-347: I'm not able to understand the meaning fo the sentences

22) Table 1 and Table 2: It is not clear what the first three columns represent. The tables show the results obtained on signatures taken from a dataset but in the text it is not explained before the references of the two tables (line 347)

23) Formula 11: missing symbol

24) Line 364: segmentation or section?

25) Line 367: How did you computer the threshold value (70)??

26) Line 408: What is the meaning of "EC are determined by cross enumeration calculation"

27) Figure 9: It is not clear what the Figure shows. Please, comment the figure better.

28) Reference: Please check all the references. In many of them a strange string of character [C]\\ or [J] appears. Furthermore, the journal of reference  13 is Pattern Recognition and the authors of reference 43 contains the symbol ?

Author Response

Please see the attachment。

Round 2

Reviewer 2 Report

The authors's answers to my previous revision are convincing. The paper can be accepted but i suggest a revision of english and style in order to make more readable the paper.

For the final revision of the paper please note that:

Please, introduce a definition of "kind" in the paper. It will help the reader to understand the method  Table I should be updated by adding the reference number 19, if i'm not in error. Be careful. In many parts of the paper you wrote "singers" instead of "signers" Symbols [c] and [j] appear again at the end of each reference. I suppose they mean for "conference paper " and "journal paper" but I suppose they are not required.

Author Response

Thank you very much for your careful review, and your comments have helped me tremendously in my thesis. I have modified the issues you mentioned and updated the writing of some of the papers. Spelling errors and references have been corrected. A definition of "kind" had been introduce in line 272.

Table I should be updated by adding the reference number 19, if i'm not in error. 

I don't quite understand what this means. I added k = 10 to the title of the table.

Thank you again for your comments on my paper and look forward to your reply.